# Effects of Key Rumen Bacteria and Microbial Metabolites on Fatty Acid Deposition in Goat Muscle

**DOI:** 10.3390/ani14223225

**Published:** 2024-11-11

**Authors:** Yan Zeng, Huilong Mou, Yongmeng He, Danping Zhang, Xiao Pan, Liping Zhou, Yujian Shen, Guangxin E

**Affiliations:** 1College of Animal Science and Technology, Southwest University, Chongqing 400715, China; zengyan@swu.edu.cn (Y.Z.); 15263348068@163.com (H.M.); yongmenghe@163.com (Y.H.); shenyj1991@126.com (Y.S.); 2Sichuan Dazhou Animal Husbandry Technology Promotion Station, Dazhou 635000, China; 13628493150@163.com; 3Hechuan Animal Husbandry Station, Chongqing 401520, China; 13628441163@163.com (X.P.); 13647642442@163.com (L.Z.)

**Keywords:** ruminant, microbes, metabolites, fatty acids

## Abstract

Metabolites and microorganisms in the rumen affect host muscle fatty acid deposition. By analyzing the correlations between ruminal microbes and metabolites and multiple fatty acids in goat muscle, we identified several ruminal metabolites and microorganisms that could potentially influence fatty acid deposition in muscle. These results provide data to support the development of targeted feeding management strategies to improve the quality of goat meat.

## 1. Introduction

The Hechuan white goat, as a local genetic resource in Chongqing, China, is a dual-purpose breed for meat and skin. It has strong adaptability, early maturity, a high slaughter rate, excellent-quality meat and skin, and a stable genetic performance. In 2023, the Hechuan white goat was successfully declared a genetic resource within national livestock and poultry. Currently, the population of Hechuan white goats is small and faces threats, such as a loss of excellent genes and performance degradation. Therefore, it is crucial to conduct an in-depth exploration of the characteristics of Hechuan white goat resources and promote their innovative utilization for the development of local animal husbandry.

The taste and texture of meat are considerably affected by fatty acids. The interplay between fatty acids and carbohydrates is widely recognized as a key factor in shaping the distinct flavor of meat, and the particular composition and types of fatty acids play a critical role in influencing flavor characteristics [1,2]. Certain essential flavor compound precursors in mutton, such as stearic acid, oleic acid, and linolenic acid, are closely associated with the intensity of mutton’s aroma, demonstrating a positive correlation between these fatty acids and the strength of mutton’s odor [3]. Furthermore, the presence of fatty acids significantly affects meat’s color, pH levels, and tenderness. Notably, the relationship between the stearic acid concentration and fat firmness affects the melting point of marbled fat and the succulence of meat [4]. In terms of the nutritional implications, the consumption of saturated fatty acids has primarily been linked to negative health effects, particularly in relation to obesity. For example, an increased intake of palmitic acid has been demonstrated to trigger an inflammatory response in the body [5]. Conversely, unsaturated fatty acids offer numerous health benefits, with oleic acid being recognized for its ability to enhance fat distribution by lowering total cholesterol and low-density lipoprotein cholesterol levels [6]. Therefore, a comprehensive understanding of the factors that influence the accumulation of fatty acids in mutton is crucial for enhancing the flavor profile and overall quality of mutton.

In recent years, increasing scholarly attention has been given to investigating the correlation between microbial composition and host productivity. Research in this area has focused on the relationship between gut flora and feed efficiency and the effect of microbial populations on the quality of dairy products [7,8]. Studies have shown that microorganisms in the rumen play a crucial role in determining the fatty acid composition of rumen digestion, particularly in terms of biohydrogenation and isomerization [9]. These processes can affect the distribution of fatty acids in different animal tissues, influencing the overall quality of animal-derived products. For example, Kim et al. identified specific bacterial taxa, such as *Verrucomicrobia*, linked to marbling in Korean beef cattle [10]. Moreover, the synthesis and deposition of fatty acids in ruminant muscles represent a complex interplay between rumen microorganisms and the host’s metabolic pathways. By utilizing metabolomics technology, Zhang et al. revealed differences in the quality of goat meat between traditionally grazed goats and their stall-fed counterparts. These differences were attributed to a reduction in unsaturated fatty acid biosynthesis in the former [11]. Ma et al. conducted a comprehensive metabolomics study that compared the flavor precursor metabolites in Hu sheep and Dorper sheep with varying levels of intramuscular fat content. They offered valuable insights into the flavor profiles of different sheep breeds based on variations in metabolite composition [12].

Limited research has explored the potential link between rumen contents and meat quality in a specific breed under consistent dietary and feeding conditions. The current study focuses on Chinese Hechuan white goats, and it aims to uncover the potential relationships between rumen microorganisms and fatty acid accumulation in muscle tissues by evaluating the correlations among rumen microorganisms and metabolites and diverse fatty acids in the longissimus dorsi muscles. The findings of this investigation offer new insights and empirical evidence to enhance the quality of goat meat.

## 2. Results and Discussion

### 2.1. Correlation Between Metabolites in Goat Rumen Contents and Fatty Acids in Their Meat

The results of fatty acid detection showed that four saturated fatty acids (myristic acid, palmitic acid, stearic acid, and heptadecanoic acid) and five unsaturated fatty acids (palmitic acid, trans-oleic acid, oleic acid, linoleic acid, and DH-γ-linolenic acid) were present in the Longissimus lumborum muscle of Hechuan white goats (Table 1).

A total of 965 metabolites were detected through a metabolomic analysis of 15 samples of goats’ rumen contents. The correlation analysis showed that 96 metabolites were significantly correlated with different fatty acids (*p* < 0.01). After conducting a weighted correlation network analysis (WGCNA), 965 metabolites were clustered into nine different-color modules (Figure 1A). Within each module were highly correlated metabolites. Following the association analysis between the modules and the traits, the results indicated that the turquoise module was significantly correlated with palmitoleic acid (R = −0.761, *p* < 0.01), stearic acid (R = 0.696, *p* < 0.01), and DH-γ-linolenic acid (R = 0.720, *p* < 0.01) (Figure 1B). The 111 metabolites in the turquoise module were compared with 96 related metabolites, and 62 metabolites that showed a high correlation with relevant fatty acids were screened (*p* < 0.05) (Figure 1F and Appendix A). In addition, we correlated the metabolites in the turquoise module with fatty acid traits separately. It was also shown that there were multiple metabolites in the module that were strongly correlated with palmitoleic acid, stearic acid, and DH-γ-linolenic acid (MM > 0.8) (Figure 1C–E).

The metabolites produced in the rumen of ruminant animals play a crucial role as necessary precursors for the synthesis of their meat and dairy products. Simultaneously, they also contain useful information on the interaction between ruminal microorganisms and diets [13]. Goat meat products exhibit considerable nutritional value, but they also contain high levels of unhealthy saturated fatty acids. The causes of these high levels are the metabolism and microbial action in the rumen of ruminants. The metabolites related to muscle fatty acids screened in this study were mostly amino acids, nucleotides and their metabolites, and organic acids. KEGG analysis showed that these metabolites were involved in metabolic pathways, ABC transporters, and the biosynthesis of cofactors (Figure 1G). Coincidentally, most of the metabolites of interest were positively correlated with stearic acid and DH-γ-linolenic acid but negatively correlated with palmitoleic acid.

This study identified a significant relationship between vitamin-B-related metabolites and the fatty acid composition in the muscle (Appendix A). For example, pantothenic acid (VB5) and D-calcium pantothenate (VB5 metabolite) were found to exhibit a significantly positive association with stearic acid content and a negative correlation with palmitoleic acid (*p* < 0.01). Pantothenate plays a key role in coenzyme A (CoA) and acyl carrier protein synthesis, influencing metabolic pathways. CoA aids in acetyl group transfer during fatty acid synthesis, leading to the conversion of palmitic acid into stearic acid with CoA’s involvement. In addition to dietary sources, pantothenic acid can be synthesized by microorganisms, such as *Escherichia coli* and *Streptococcus bovis*, in ruminants’ rumens [14]. Niacin (VB3) exhibited a positive correlation with stearic acid, while 6-methylnicotinamide (a VB3 metabolite) presented a positive correlation with DH-γ-linolenic acid and a negative correlation with palmitoleic acid. Niacin is crucial for the conversion of the volatile fatty acids produced by ruminants, such as acetic acid, into long-chain fatty acids [15]. It serves as a precursor for NADPH, a key enzyme in the de novo synthesis of fatty acids [16]. Additionally, niacin is recognized for its anti-lipolytic properties, which operate through a hydroxycarboxylic acid-2-receptor-dependent mechanism [17]. The incorporation of niacin into animal feed has a notable impact on their intermuscular fat content and the marbling score of beef [18]. This conclusion was also obtained in a study on Chinese Tan sheep [12]. Moreover, niacin affects the composition of ruminal microorganisms to some degree, such as suppressing starch utilization; promoting fiber degradation by decreasing the populations of *Proteobacteria*, *Succiniclasticum*, and *Treponema*; and enhancing the prevalence of *Prevotella* [17]. Currently, research on using vitamin B as a feed additive for the nutrition of beef cattle or pigs is relatively abundant, whereas reports concerning its effects on improving meat quality in goats remain scarce [19,20,21].

### 2.2. Correlation Between Rumen Microorganisms and Fatty Acid Content in the Dorsal Longissimus Lumborum Muscles of Hechuan White Goats

The rumen of ruminants plays a pivotal role in natural fermentation, characterized by a substantial population and diverse array of microorganisms. Fatty acids in ruminants are predominantly synthesized through the metabolic activities of rumen microbes, alongside exogenous absorption. Notably, butyrate, a product of microbial fermentation, is absorbed by epithelial cells in the rumen and subsequently converted into β-hydroxybutyrate, acetate, and other short- and medium-chain fatty acids [22]. Conversely, long-chain fatty acids are primarily derived from dietary lipids. Ruminant meat exhibits high levels of saturated fatty acids. This condition is a consequence of microbial hydrogenation within the rumen ecosystem. An analysis of the rumen microorganisms in goats revealed a consistent species composition across different individuals. At the phylum level, Bacteroidetes and Firmicutes emerged as prominent species, although variations in the abundance ratios were observed among individuals. *Prevotella* and *Bacteroides* were identified as the principal species at the genus level, affirming the findings of previous investigations [23].

To explore the bacterial species that are strongly related to fatty acids further, we grouped and analyzed individuals with different fatty acid contents (Figure 2).

The abundance of *Eubacterium ruminantium* and *Fibrobacter succinogenes* in the rumen of the high palmitoleic acid group was significantly greater than that in the low palmitoleic acid group, with a difference exceeding 1% (*p* < 0.05) (Figure 2A). Notably, the abundance of *Eubacterium ruminantium* reached its highest level in the high palmitoleic acid group, at 3.19%. *Eubacterium ruminantium* is recognized as a significant fiber-dissolving bacterial species [24,25]. Research has indicated that following carbohydrate ingestion, the gastrointestinal microbiota, including *Eubacterium ruminantium*, catabolize carbohydrates and generate unsaturated fatty acids, such as palmitoleic acid [26]. For beef cattle, the abundance of *Eubacterium ruminantium* and *Fibrobacter succinogenes* is positively correlated with feed efficiency [27]. This suggests that the meat quality of Hechuan white goats can be controlled by regulating the abundance of *Eubacterium ruminantium* in the rumen of these goats during feeding.

Between the high and low stearic acid content groups, the abundance of cellulolytic bacteria, *Prevotella P2-180*, and *Prevotella P5-126* was greater than 0.01% and significantly different (*p* < 0.05) (Figure 2A). *Prevotella* is one of the most abundant bacterial genera found in the rumen, and these bacteria break down cellulose as an energy source in goats [28]. Studies in Angus cattle have shown a positive correlation between *Prevotella* and intramuscular fat content [9]. This suggests that *Prevotella* may be capable of converting stearic acid in the body into intramuscular fat, which is deposited in the muscle, resulting in the meat having a better taste. Simultaneously, *Prevotella* regulates glucose metabolism by promoting glycogen storage and stimulating the expression of the glucagon signaling pathway. The KEGG annotation results based on the difference in functional abundance between the high and low stearic acid groups also showed that the glucagon signaling pathway was significantly different in the high stearic acid group (Figure 2B). The correlation analysis between microbial species abundance and key metabolites also indicated that *Prevotella* was negatively correlated with a variety of metabolites that were highly correlated with stearic acid content, such as methylguanosine and p-coumaric acid (Figure 2C). Between the high and low DH-γ-linolenic acid content groups, the difference in the presence of *Peptostreptococcaceae P7* in the rumen was significant (*p* < 0.05). *Peptostreptococcaceae* bacteria in the rumen are among the most important cellulose- and hemicellulose-degrading bacteria in goats, and they play a crucial role in fiber degradation. Several studies have found that *Peptostreptococcaceae bacteria* in the rumen not only affect fat deposition but also inhibit weight gain [29]. This finding suggests that paying attention to changes in the abundance of *Peptostreptococcaceae bacteria* in the rumen of Hechuan white mountain goats is necessary during production. A high abundance of *Peptostreptococcaceae bacteria* in the rumen may lead to a decrease in the feed conversion rate and affect the economic efficiency of a farm.

## 3. Materials and Methods

### 3.1. Experimental Animals

A total of 15 6-month-old healthy male Hechuan white goats were selected for this study. All the goats were bred and fed under the same conditions. All the experimental goats were slaughtered in accordance with the requirements of the Health and Quarantine Code for Slaughtering Livestock and Poultry (NY 467-2001) [30] and the standard of the Sheep Slaughtering Operation Rules (GB/T 43562-2023) [31]. The goats were slaughtered after 24 h with food deprived and had free access to water. Before slaughter, goats should be stunned with an electric shock. The esophagus, trachea, and carotid arteries were transected in the larynx, and the time interval between stunning and slaughter was less than 1 min. Subsequently, the longissimus dorsi between the 12 and 13th rib was excised from all 15 goats on the left side of each carcass within 30 min after exsanguination.

### 3.2. Metabolome Analysis of Rumen Contents and Fatty Acid Composition of the Longissimus Dorsi in Goats

In this step, 50 mL of homogenized rumen contents was collected individually from each goat for analysis. An ultra-high-performance liquid chromatography–tandem mass spectrometry platform was used to detect the metabolites in a portion of the rumen contents. Weighted gene co-expression network analysis (WGCNA) was conducted to cluster the detected metabolites, and the correlation between the clustered modules and the results on the fatty acid composition in the meat was analyzed. Pearson’s correlation coefficients were calculated in R to determine the associations between metabolites and fatty acid traits. A correlation coefficient greater than or equal to an absolute value of 0.6 (i.e., r ≥ ±0.6) was used as the threshold for reporting because correlation coefficients larger than ±0.60 are considered of moderate or higher strength. Metabolites with a strong correlation were subjected to Kyoto Encyclopedia of Genes and Genomes (KEGG) enrichment analysis.

### 3.3. Correlation Analysis Between Microbial Metagenomes in the Rumen and Fatty Acids in the Muscle in Goats

Genomic DNA was extracted from the samples using the HiPure bacterial DNA kit (Guangzhou, China). Sequencing libraries were constructed using the NEBNext^®^ΜLtraTMDNA Library Prep Kit (NEB, Ipswich, MA, USA), and sequencing was performed on an Illumina Novaseq 6000 sequencer. Raw data from the Illumina platform were filtered using FASTP (version 0.18.0). Contigs were generated via de novo assembly of clean reads from the samples using MEGAHIT (V1.1.2) software. MetaGeneMark (version 3.38) was used for comparison, and a 500 bp contig was used for the gene prediction. CD-HIT was used to cluster the DNA sequences to generate nonredundant sequences. Pathoscope (V2.0.7) was used to calculate the gene abundance based on the results of the read alignment for each sample output by Bowtie software. DIAMOND (V0.9.24) was used to align the nonredundant gene sets with the KEGG database for annotation. Functional abundance was calculated based on the number of genes in the annotated pathways. The fatty acids that exhibited a high correlation with the metabolome in the nutritional composition test results for the meat were selected, and the high and low groups were set as shown in supplementary material (Appendix A). Kaiju (V1.6.3) was used to compare the Nr (RefSeq nonredundant protein database) database for species annotation at each classification level. Welch’s *t*-test was used to compare the mean differences in species/function between the two groups using the Vegan package in the R language. A *p* value of <0.05 was the threshold of significance.

## 4. Conclusions

Under the same dietary structure and feeding environment, differences in rumen microorganisms and metabolism will exert certain effects on production traits. Here, the present results show that dietary vitamin B group components exhibited a correlation with the composition of fatty acids in the muscle. In addition, the abundance of *Bacteroides*, *Ruminococcaceae P7*, *Eubacterium ruminantium*, and *Prevotella* in the rumen was significantly correlated with fatty acid content. Therefore, by improving feeding management—such as regulating the niacin content in the diet or adding corresponding probiotics—the abundance of related bacteria in the rumen can be regulated, the stearic acid content in the meat can be moderately reduced, and the flavor of goat meat can be improved. These results provide data support for the development of targeted feeding management strategies to improve the quality of goat meat.

## Figures and Tables

**Figure 1 animals-14-03225-f001:**
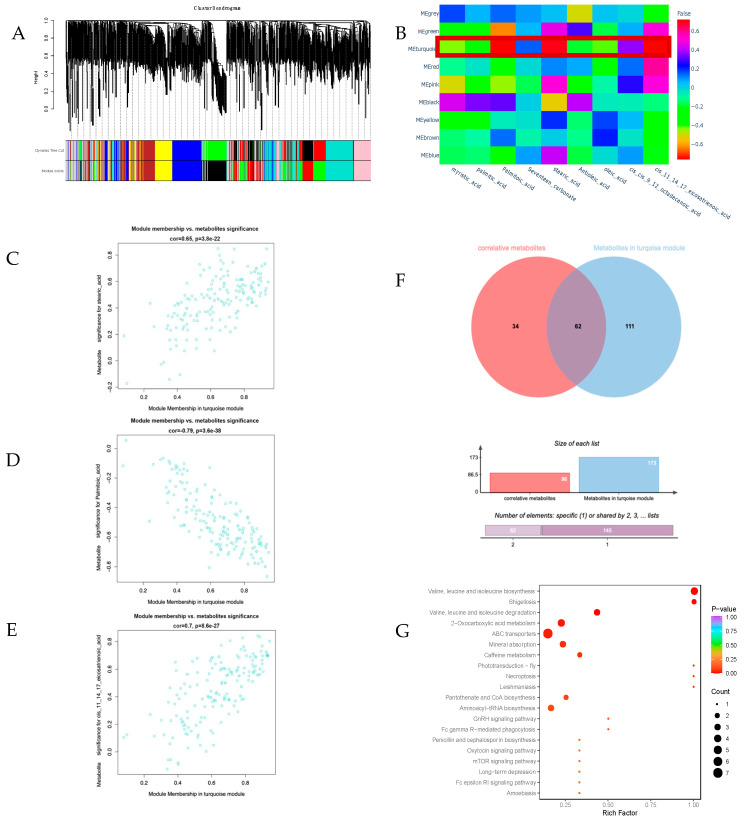
Correlation between metabolites in goat rumen contents and fatty acids in meat. (**A**) Cluster map of the rumen contents of Hechuan white goat metabolome modules. (**B**) Module–fatty acid trait correlation diagram; each row represents a module, and each column represents a fatty acid trait. (**C**) Scatter plot of the significance (MS) of the metabolites in the module membership (MM) versus stearic acid in the turquoise module. (**D**) Scatter plot of palmitoleic acid versus the significance (MS) of metabolites in the module membership (MM) in the turquoise module. (**E**) Scatter plot of DH-γ-linolenic acid versus the significance (MS) of metabolites in module membership (MM) in the turquoise module. (**F**) Venn plot of metabolites significantly associated with the metabolite set in the turquoise module. (**G**) Bubble plot of KEGG pathway enrichment analysis for metabolites of interest.

**Figure 2 animals-14-03225-f002:**
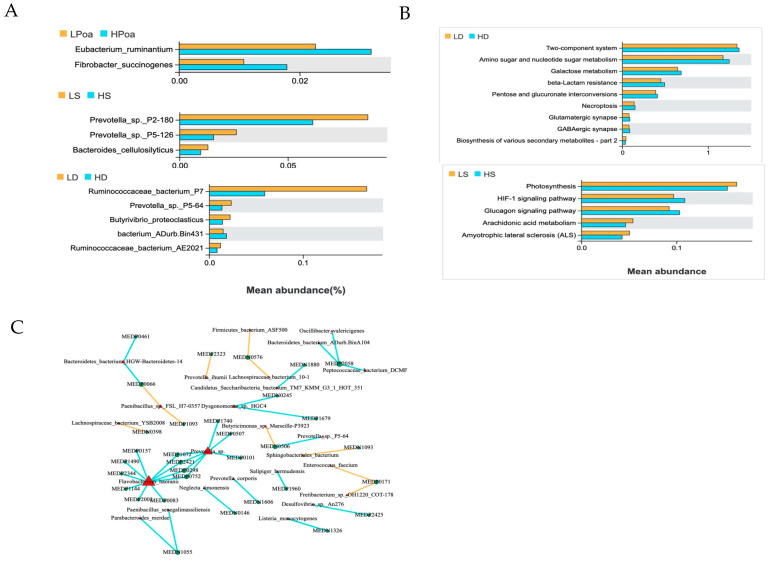
(**A**) Bacterial species exhibiting significant differences between groups and with an abundance greater than 1% (*p* < 0.05); (**B**) KEGG annotation pathway with significant differences between groups; (**C**) network of correlation analysis between microbial species abundance and key metabolites. Lpoa: low palmitoic acid group; Hpoa: high palmitoic acid group; LS: low stearic acid group; HS: high stearic acid group; LD: high Dh-γ-linolenic acid; HD: high Dh-γ-linolenic acid group.

**Table 1 animals-14-03225-t001:** Metabolome clustering module results for the correlation of rumen contents with fatty acid content.

Fatty Acid	Maximum	Minimum	Mean ± SD
Myristic acid	5.42	1.83	3.46 ± 1.17
Palmitic acid	27.4	21.4	24.77 ± 1.86
Palmitoleic acid	2.88	1.47	2.13 ± 0.48
C17:0	1.79	1.31	1.51 ± 0.18
Stearic acid	22	15	17.63 ± 2.23
Elaidic acid	4.35	2.27	3.01 ± 0.64
Oleic acid	46	33.9	41.01 ± 3.33
Linoleic acid	9.04	3.53	5.52 ± 1.64
Dh-γ-linolenic acid	5.63	1.49	2.83 ± 1.33

## Data Availability

The data are contained within the article and Appendix A.

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
