# Peer review of "Effects of Key Rumen Bacteria and Microbial Metabolites on Fatty Acid Deposition in Goat Muscle"

_animals, 2024, doi:10.3390/ani14223225_

Round 1
Reviewer 1 Report
Comments and Suggestions for Authors
-
Expand on the Mechanisms: While the correlation between vitamin B-related metabolites and fatty acid deposition is established, a deeper exploration into the biological mechanisms behind these correlations could strengthen the study's impact.
-
Longitudinal Data: If possible, including longitudinal data on the changes in rumen microbiota and muscle fatty acid composition over time could provide a more dynamic view of the relationships being studied.
-
Diet Variation: Although the study controls for diet structure, exploring how different dietary components might interact with the rumen microbiome to influence fatty acid deposition could offer additional insights.
-
Comparative Analysis: Comparing the findings with other ruminant species could provide a broader context and highlight the uniqueness or commonality of the observed effects in goats.
-
Implications for Industry: A discussion on how these findings could be implemented in the livestock industry to improve meat quality and potentially increase market value would be beneficial.
- Article Type Change: I would like to sincerely suggest that the authors expand the content of the introduction and discussion, increase the number of references to more than 30, and change the type of paper to an Article. This is because papers of the Communication type are not always recognized, especially in China.
Author Response
Comments1: Expand on the Mechanisms: While the correlation between vitamin B-related metabolites and fatty acid deposition is established, a deeper exploration into the biological mechanisms behind these correlations could strengthen the study's impact.
Response1:Thanks for your suggestions, we have made appropriate changes in the manuscript.
Line 136-line 148: Niacin is crucial for the conversion of volatile fatty acids produced by ruminants, such as acetic acid, into long-chain fatty acids [15]. It serves as a precursor for NADPH, a key enzyme in the de novo synthesis of fatty acids [16]. Additionally, niacin is recognized for its anti-lipolytic properties, which operate through a hydroxycarboxylic acid-2-receptor-dependent mechanism [18]. The incorporation of niacin in animal feed has a notable impact on intermuscular fat content and the marbling score of beef [17]. Moreo-ver, niacin affects the composition of ruminal microorganisms to some degree, such as suppressing starch utilization, promoting fiber degradation by decreasing the popula-tions of Proteobacteria, Succiniclasticum, and Treponema, and enhancing the preva-lence of Prevotella [18]
Comments2:Longitudinal Data: If possible, including longitudinal data on the changes in rumen microbiota and muscle fatty acid composition over time could provide a more dynamic view of the relationships being studied.
Response2: Thank you for your suggestion. The longitudinal collection of data over time, particularly for muscular characteristics, may necessitate the selection of different batches of goat samples. Given that the foraging behavior of goats is significantly influenced by seasonal and climatic variations, discrepancies in data among different batches could introduce substantial errors. Therefore, this study was conducted using only a single time point for analysis.
Comments3:Diet Variation: Although the study controls for diet structure, exploring how different dietary components might interact with the rumen microbiome to influence fatty acid deposition could offer additional insights.
Response3:Thank you for your suggestion. The objective of this study is to investigate the correlation between fatty acid deposition in muscle and the contents of the rumen. We aim to identify metabolites and microorganisms associated with fatty acids, which may provide guidance for subsequent nutritional feeding strategies. Further research will continue to explore how various dietary components interact with the rumen microbiome.
Comments4:Comparative Analysis: Comparing the findings with other ruminant species could provide a broader context and highlight the uniqueness or commonality of the observed effects in goats.
Response4: Thank you for your suggestionsWe conducted a comparative analysis of specific data with respect to cattle and sheep presented in the manuscript and incorporated suitable supplements.
Line171-175 Research has indicated that following carbohydrate ingestion, gastrointestinal microbio-ta, including Eubacterium ruminantium, catabolize carbohydrates and generate un-saturated fatty acids, such as palmitoleic acid [27]. For beef cattle, the abundance of Eu-bacterium ruminantium and Fibrobacter succinogenes is positively correlated with feed efficiency [28].
Line182-184:Studies Angus cattle have shown a positive correlation between Prevotella and intramuscular fat content [9]. This suggests that Prevotella may be capable of converting stearic acid in the body into intramuscular fat, which is deposited in the muscle, resulting in a better taste of meat.
Comments5: Implications for Industry: A discussion on how these findings could be implemented in the livestock industry to improve meat quality and potentially increase market value would be beneficial.
Response5: Thank you for your suggestions, we have made appropriate changes in the manuscript.
Line254-270:Therefore, by improving feeding management—such as regulating the niacin content in the diet or adding corresponding probiotics—the abundance of related bacteria in the rumen can be regulated, the stearic acid content in the meat can be moderately re-duced, and the flavor of goat meat can be improved. These results provide data sup-port for the development of targeted feeding management strategies to improve goat meat quality.
Comments6: Article Type Change: I would like to sincerely suggest that the authors expand the content of the introduction and discussion, increase the number of references to more than 30, and change the type of paper to an Article. This is because papers of the Communication type are not always recognized, especially in China.
Response6: Thanks for your suggestion. References have been added through the supplement and revision of the manuscript; however, due to the format of the data (mostly tables, in the attachment), we still present it in the Communication mode.
Reviewer 2 Report
Comments and Suggestions for Authors
Animals-3287648 investigated the relationship between meat fatty acid composition, ruminal microbiota, and metabolome in goats. It is noteworthy that the study clarified the relationship between B vitamins, certain ruminal microorganisms and the fatty acid composition of meat.
General comment
I think that detailed explanations of figures and tables make it easier for readers to understand the value of the paper.
Specific comments
L85: What is Iborum muscle?
Table1: It is recommended to identify compounds in C17:0 if possible.
L92 others: I would recommend that you explain in detail in the text what the Turquoise Module is and what it means.
L94: Please do not summarize Figures 1B-E, but show the results individually. Please carefully show the correspondence between the figures and the data.
Figure1B: Please explain in detail what the vertical axis means.
Figure1C,D,E: Please explain in detail what the numbers on the vertical axis mean.
Figure1: Please provide detailed information in the footnotes for Figure 1.
L118-120 Which chart shows these results?
L176-177 Which chart shows these results?
Figure2A: What is “case”?
L188-189: Which are the results of LS and HS?
Author Response
Comments 1: L85: What is Iborum muscle?
Response 1: longissimus lborum muscle:The longissimus muscle is a deep muscle of the back that spans the entire length of the vertebral column. It assumes a central position within the erector spinae group, in between the spinalis and iliocostalis muscles.
Comments 2: Table1: It is recommended to identify compounds in C17:0 if possible.
Response 2: Thank you for your suggestion. Trace amounts of Seventeen carbonate were detected in only a subset of samples. Further identification of its components could not be achieved.
Comments 3: L92 others: I would recommend that you explain in detail in the text what the Turquoise Module is and what it means.
Response 3: Thank you for your suggestion. After correlation analysis of the within-module and traits, only the turquoise module had a high correlation with the phenotype (R>0.6). Therefore, we further narrowed the range of target metabolites (96 to 62) by crossing the metabolites in the turquoise module with the 96 relevant metabolites obtained in the previous stage(fig 1F).
Comments 4: L94: Please do not summarize Figures 1B-E, but show the results individually. Please carefully show the correspondence between the figures and the data.
Response 4: Thank you for your suggestion. We did have some unclear statements in the manuscript and the figure, which have been corrected in the manuscript.
A total of 965 metabolites were detected through the metabolomic analysis of 15 goat rumen content samples. Correlation analysis showed that 96 metabolites were sig-nificantly correlated with different fatty acids (p < 0.01). After conducting a weighted cor-relation network analysis (WGCNA), 965 metabolites were clustered into nine different col-or modules (Figure 1A). Within each module are highly correlated metabolites. Follow-ing the association analysis between the modules and traits, the results indicated that the turquoise module was significantly correlated with palmitoleic acid (R = −0.761, p < 0.01), stearic acid (R = 0.696, p < 0.01), and DH-γ-linolenic acid (R = 0.720, p < 0.01) (Figure 1B). The 111 metabolites in the turquoise module were compared with 96 relat-ed metabolites, and 62 metabolites that showed a high with relevant fatty acids were screened (p < 0.05) (Figure 1F, Table S2). In addition, we correlated the metabolites in the turquoise module with fatty acid traits separately. It was also shown that there were multiple metabolites in the module that were strongly correlated with palmitoleic acid, stearic acid, and DH-γ-linolenic acid (MM > 0.8) (Figure. C-E).
Comments 5: Figure1B: Please explain in detail what the vertical axis means.
Response 5: Weighted correlation network analysis (WGCNA) was used for cluster analysis according to the expression pattern of metabolites. All metabolites were clustered into 9 different color modules (Figure 1A). Within each module are highly correlated metabolites. The correlation analysis between modules and traits was performed to find out the module with the highest correlation with the concerned traits (Figure 1B). The vertical axis represents the module.
Comments 6: Figure1C, D, E: Please explain in detail what the numbers on the vertical axis mean.
Response 6: The vertical axis represents the degree of correlation between metabolites and traits, with higher values indicating a higher degree of correlation with traits.
Comments 7: Figure1: Please provide detailed information in the footnotes for Figure 1.
Response 7: Thanks for your suggestion, we have made changes in the manuscript
Figure 1: Correlation between metabolites in goat rumen content and fatty acids in meat. A: Cluster map of the rumen contents of Hechuan White goat metabolome modules. B: Module-fatty acid trait correlation diagram; each row represents a module, and each column represents a fatty acid trait. C: Scatter plot of the significance (MS) of the metabolites in the module members (MM) versus stearic acid in the turquoise module. D: Scatter plot of palmitoleic acid versus the signifi-cance (MS) of metabolites in the module members (MM) in the turquoise module. E: Scatter plot of DH-γ-linolenic acid versus the significance (MS) of metabolites in module members (MM) in the turmeric module. F: Venn plot of metabolites significantly associated with the metabolite set in the turquoise module. G: Bubble plot of KEGG pathway enrichment analysis for metabolites of in-terest.
Comments 8: L118-120 Which chart shows these results?
Response 8: Thank you for your suggestion. I apologize for not being clear. The relevant results, constrained by the space available in the table, are all present in the attached table.
Comments 9: L176-177 Which chart shows these results?
Response 9: Thank you for your reminder, which we have incorporated into the manuscript.
Comments 10-11: Figure2A: What is “case”? L188-189: Which are the results of LS and HS?
Response 10-11: Thank you for your reminder, I apologize for our mistake. The case is HS, and the control is LS, which has been changed in the manuscript.
Round 2
Reviewer 1 Report
Comments and Suggestions for Authors
The authors have addressed my concerns, and I think it is ready for publication. Good Luck!